# High-Entropy Oxides: Advanced Research on Electrical Properties

**Haoyang Li [1], Yue Zhou [1], Zhihao Liang [1], Honglong Ning [1,\*] , Xiao Fu [1], Zhuohui Xu [2], Tian Qiu [3], Wei Xu [1], Rihui Yao [1,\*] and Junbiao Peng [1]**

1   Institute of Polymer Optoelectronic Materials and Devices, State Key Laboratory of Luminescent Materials and Devices, South China University of Technology, Guangzhou 510640, China; 201830640231@mail.scut.edu.cn (H.L.); 201730341030@mail.scut.edu.cn (Y.Z.); 201530291443@mail.scut.edu.cn (Z.L.); 201630343721@mail.scut.edu.cn (X.F.); xuwei@scut.edu.cn (W.X.); psjbpeng@scut.edu.cn (J.P.)

2   Guangxi Key Lab of Agricultural Resources Chemistry and Biotechnology, Yulin Normal University, Yulin 537000, China; xzh21@ylu.edu.cn

3   Department of Intelligent Manufacturing, Wuyi University, Jiangmen 529020, China; qiutian@ustc.edu

\*   Correspondence: ninghl@scut.edu.cn (H.N.); yaorihui@scut.edu.cn (R.Y.)

**Abstract:** The concept of "high entropy" was first proposed while exploring the unknown center of the metal alloy phase diagram, and then expanded to oxides. The colossal dielectric constant found on the bulk high-entropy oxides (HEOs) reveals the potential application of the high-entropy oxides in the dielectric aspects. Despite the fact that known HEO thin films have not been reported in the field of dielectric properties so far, with the high-entropy effects and theoretical guidance of high entropy, it is predictable that they will be discovered. Currently, researchers are verifying that appropriately increasing the oxygen content in the oxide, raising the temperature and raising the pressure during preparation have an obvious influence on thin films' resistivity, which may be the guidance on obtaining an HEO film large dielectric constant. Finally, it could composite a metal–insulator–metal capacitor, and contribute to sensors and energy storage devices' development; alternatively, it could be put into application in emerging thin-film transistor technologies, such as those based on amorphous metal oxide semiconductors, semiconducting carbon nanotubes, and organic semiconductors.

**Keywords:** high-entropy oxides; thin films; dielectric constant; oxygen content





## 1. Introduction

In order to further explore the central region of the multi-component metal phase diagram (Figure 1) in metallurgy [1], Cantor [2] et al. found that, under Gibbs phase rule, approximately equimolar mixing could yield the permitted number of phases, even for single-phase alloys. Then, Murty et al. [3] explicitly defined the concept of high-entropy alloys (HEAs), that is, alloys containing at least five principal elements, each having an atomic percentage between 5% and 35%. Due to its high hardness [4,5], high stability [6], high wear resistance [7,8], oxidation resistance [9,10] and other excellent properties, HEAs have attracted worldwide scientists' attention. So far, research on HEAs and theoretical systems has been relatively complete. Inspired by the remarkable properties of them, researchers speculated that oxides could also exhibit better performance under the influence of high entropy [11]. The first time applying the concept of high entropy to oxides was in 2015, when Rost et al. [12] successfully prepared an entropy-stabilized crystal (MgCoNiCuZn)O by filling a single sublattice with five different cations, and first proposed the concept of "Entropy-Stabilized Oxides". Bérardan et al. [13,14] followed up with a study of its electrical conductivity and dielectric coefficient and renamed it "high entropy oxides" (HEOs). Since then, more types of remarkable properties have been

discovered while more scientists conduct research on them. Due to the impacts of high entropy on thermodynamics and structure, the properties of HEOs are different from those of ordinary oxides. Known excellent properties range from magnetic properties [15,16] to electrochemical properties [17,18], from catalytic properties [19] to thermal conductive properties [20], etc. In particular, HEOs' electrical properties, specifically the dielectric property, attract scientists' attention. It is reported that it can achieve a colossal dielectric constant, higher than 4800 [13]. As the trend towards the miniaturization of today's and future electronic devices brings an increase in the tunneling current, materials with high permittivity are potentially of great value. Therefore, HEOs with a colossal dielectric constant satisfy recent electronic devices' requirements and could be the key to solving the current bottleneck. Moreover, the mixing of various elements allows for the design of more materials with distinctive properties, and its variety and flexibility meet modern materials' requirements. Significantly, many HEOs are based on magnetic transition metal ions; this is because electrical and magnetic properties of ferrite or manganite are strongly related to their bond lengths and bond angles, which can be easily changed by paramagnetic or diamagnetic cation substitutions [21,22]. At the same time, the disorder of the system can be further increased by controlling the concentration of doping cations and their oxidation state [23], which makes it easier for researchers to regulate HEOs' properties according to the demand. However, the fabrication temperature of HEOs is usually high, and the mechanism behind it is not yet clear, so it is necessary to conduct more studies in order to effectively guide the design and preparation of materials.

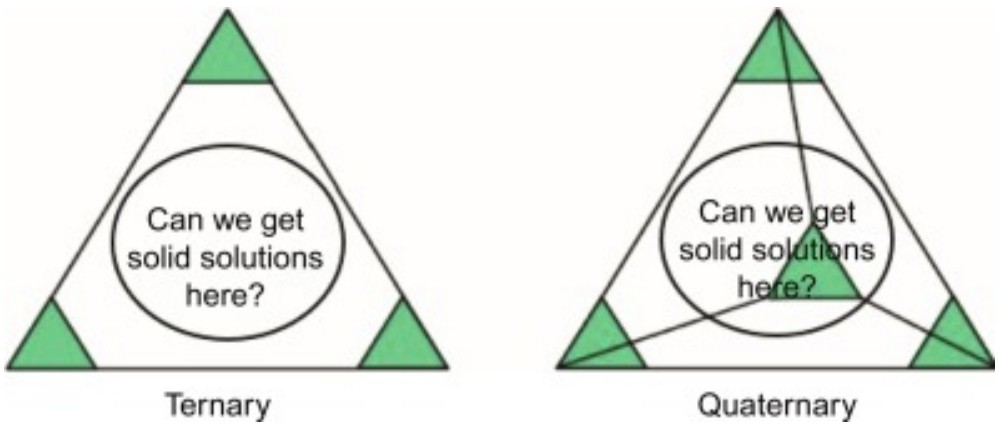

**Figure 1.** Sketch of metal phase diagram [1].

In our study, we first introduce the definition of high entropy and roughly classify HEOs according to elements or structures. Then, we discuss the electrical properties of HEOs, followed with that of HEO films. Remarkably, HEO films are usually deposited on a monocrystalline silicon sheet or a glass substrate by different sputtering means in the oxygen atmosphere. Finally, we advance future potential applications as gate dielectrics after presenting current mature applications.

## 2. High Entropy

In statistical physics, according to the Boltzmann formula and the additivity property of entropy, we can calculate the configurational entropy(S) by Formula (1) [24,25]:

$$S = -R[x_A lnx_A + x_B lnx_B + \cdots] \tag{1}$$

where $R$ is the molar gas constant and $x_A$ represents the proportion of $A$ atoms in all atoms. Since the configurational entropy of oxygen ions in an ideal solid solution approximately equals zero, the whole crystal entropy is approximately the sum of cation entropy. Thus, we know that in order to maximize the S value, the ions should have the same proportion, that is, equal molar composition. When five components are equally mixed, S > 1.5$R$,

which is high enough compared with the ordinary materials' entropy (for example, the configuration entropy of liquid stainless steel is 0.96*R*, and even the higher value of nickel-based alloy is only 1.37*R* [26]). Murty et al. [27] used 1*R* and 1.5*R* as boundaries and divided materials into three categories: low entropy (<1*R*), medium entropy (1*R*~1.5*R*), and high entropy (>1.5*R*), as shown in Figure 2.

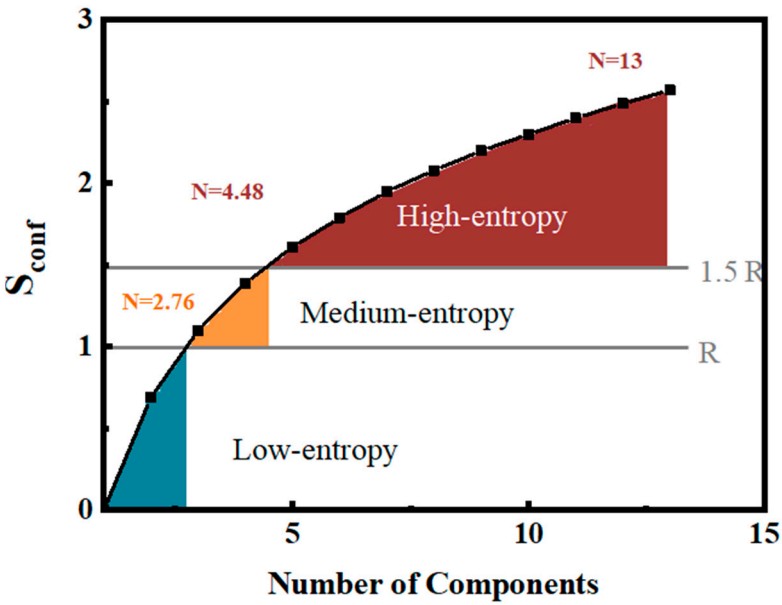

**Figure 2.** Entropy as a function of number of components.

Therefore, it is generally believed that HEOs are single-phase oxides composed of five or more kinds of cations. When there are more than five sorts of cations, the proportion is not strictly in accordance with the equimolar ratio but can fluctuate from 5 to 35% [28]. In particular, for oxides, the configurational entropy usually only considers cations, that is, oxygen atoms are not included, while for alloys, the configurational entropy includes all constituent elements. The consequent problem is that the oxide's configurational entropy is more complex than that of the alloy, because there may be a situation such as cationic site inversion in the spinel structure, which increases the difficulty of calculating configurational entropy accurately.

Meanwhile, according to Gibbs free energy (Formula (2)):

$$\Delta G_{mix} = \Delta H_{mix} - T\Delta S_{mix} \tag{2}$$

where $\Delta G_{mix}$ is the change of Gibbs free energy for the system; $\Delta H_{mix}$ is the enthalpy change for the system; $\Delta S_{mix}$ is the entropy change for the system; T is the absolute temperature. It can be deduced that a high temperature is conducive to the synthesis and stability of HEOs. The Gibbs free energy of HEOs tends to be negative at high temperatures, usually around 1000 °C (900 °C [29]~1400 °C [30]). More interestingly, the effect of configuration entropy is even greater for oxides. For instance, the removal of any element in (MgCoNiCuZn)O would lead to the failure of single-phase formation [12], while taking the same measure for HEA CrMnFeCoNi will not result in a failure single-phase solid solution state [31]. This indicates that alloy's medium entropy might counteract its enthalpy change, but the medium entropy of the oxide does not.

It is known that properties of oxides are largely related to their structures. Moreover, high entropy brings four effects: 1. high-entropy effect on thermodynamics [24]; 2. lattice distortion effect on the structure [32] (Figure 3); 3. sluggish diffusion effect [33]; 4. cocktail effect [34].

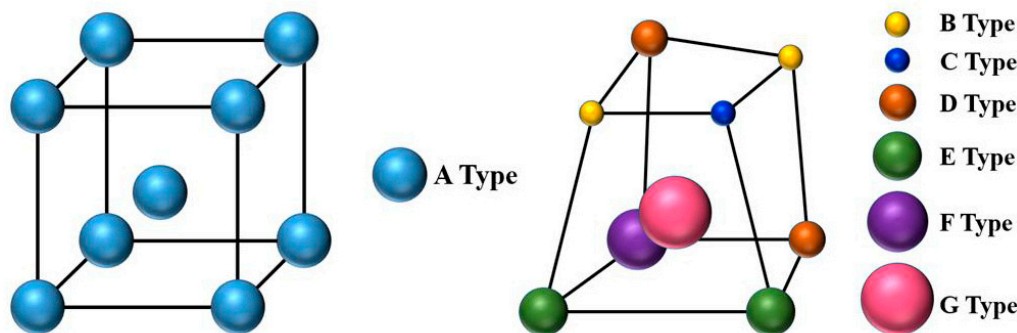

**Figure 3.** Schematic illustration of large lattice distortion that exists in the BCC lattice.

Those effects, together with structure, largely determine the properties of HEOs, such as dielectric, electrochemical, optical, magnetic, mechanical, etc. In this review, we mainly analyze electrical properties.

### 3. Bulk High-Entropy Oxides

*3.1. Classification*

According to reported literature, HEOs can be divided into transition metal oxides and rare earth metal oxides in terms of elements. Transition metals usually refer to the 48 elements of 10 groups from Group III to Group XII in the periodic table of chemical elements, but not including the f region. The rare earth elements include 17 elements of lanthanide, plus yttrium and scandium. It can be seen from Table 1 that the present study focuses on the (MgCoNiCuZn)O system; this is because it is the first reported HEO. Due to the high price of rare earth elements, considering their future development, more research is currently focused on transition metal oxide systems, and even rare earth metal oxides are only doped with one or two elements. Interestingly, according to the table we sorted out, so far, scientists only attempted to add rare earth elements into the perovskite and fluorite structures.

Although there are various types of cations involved in HEOs, their crystal structures, however, are simple and regular, which is very convenient for scientists to study. So far, according to known studies, the crystal structures (Figure 4) include rock salt (sodium chloride crystal) represented by (MgCoNiCuZn)O [35–37], $AB_2O_4$ type structure (spinel) represented by $(CoCrFeMnNi)_3O_4$ [16,38], $ABO_3$ type structure (perovskite) represented by $La(Co_{0.2}Cr_{0.2}Fe_{0.2}Mn_{0.2}Ni_{0.2})O_3$ [15,39], and fluorite structure represented by $(Gd_{0.2}La_{0.2}Ce_{0.2}Hf_{0.2}Zr_{0.2})O_2$ [40,41], etc. Most of them are face-centered cubic (BCC) or body-centered cubic (FCC) unit cells.

**Table 1.** Four main types of crystal structures with different compositions.

| Crystal Structure | Composition | Preparation Temperature | Ref No | Year |
|---|---|---|---|---|
| Rock salt | (MgCoNiCuZn)O | 850–900 °C | [12] | 2015 |
| | (MgCoNiCuZn)LiO | 1000 °C | [14] | 2016 |
| | (MgCoNiCuZn)LiGaO | 1000 °C | [14] | 2016 |
| | PtMgCoNiCuZnO$_x$ | 900 °C | [19] | 2018 |
| | MgCoNiCuZnScO | 800–1000 °C | [42,43] | 2019 |
| | MgCoNiCuZnSbO | 800–1000 °C | [42,43] | 2019 |
| | MgCoNiCuZnSnO | 800–1000 °C | [42,43] | 2019 |
| | MgCoNiCuZnCrO | 800–1000 °C | [42,43] | 2019 |
| | MgCoNiCuZnGeO | 800–1000 °C | [42,43] | 2019 |

**Table 1.** *Cont.*

| Crystal Structure | Composition | Preparation Temperature | Ref No | Year |
|---|---|---|---|---|
| Spinel | $(Cr_{0.2}Mn_{0.2}Fe_{0.2}Co_{0.2}Ni_{0.2})_3O_4$ | 1050 °C | [44] | 2017 |
| | $(Mg_{0.2}Fe_{0.2}Co_{0.2}Ni_{0.2}Cu_{0.2})Fe_2O_4$ | 1250 °C | [45] | 2019 |
| | $(Mg_{0.2}Co_{0.2}Ni_{0.2}Cu_{0.2}Zn_{0.2})Fe_2O_4$ | 1250 °C | [45] | 2019 |
| | $(Mg_{0.2}Mn_{0.2}Co_{0.2}Ni_{0.2}Cu_{0.2})Fe_2O_4$ | 1250 °C | [45] | 2019 |
| | $(Mn_{0.2}Fe_{0.2}Co_{0.2}Ni_{0.2}Cu_{0.2})Fe_2O_4$ | 1250 °C | [45] | 2019 |
| | $(Mn_{0.2}Fe_{0.2}Co_{0.2}Ni_{0.2}Cu_{0.2})_3O_4$ | 1250 °C | [45] | 2017 |
| | $(Mg_{0.2}Fe_{0.2}Co_{0.2}Ni_{0.2}Cu_{0.2})_3O_4$ | 1250 °C | [45] | 2019 |
| | $(Mg_{0.2}Co_{0.2}Ni_{0.2}Cu_{0.2}Zn_{0.2})Al_2O_4$ | 1600 °C | [45] | 2019 |
| | $(Mg_{0.2}Co_{0.2}Ni_{0.2}Cu_{0.2}Zn_{0.2})Cr_2O_4$ | 1250 °C | [45] | 2019 |
| | $(Mg_{0.2}Fe_{0.2}Co_{0.2}Ni_{0.2}Cu_{0.2})Cr_2O_4$ | 1250 °C | [45] | 2019 |
| | $(Mg_{0.2}Mn_{0.2}Co_{0.2}Ni_{0.2}Cu_{0.2})Cr_2O_4$ | 1250 °C | [45] | 2019 |
| | $(Mn_{0.2}Fe_{0.2}Co_{0.2}Ni_{0.2}Cu_{0.2})Cr_2O_4$ | 1250 °C | [45] | 2019 |
| | $(ZnFeNiMgCd)Fe_2O_4$ | 520 °C | [46] | 2019 |
| | $(Cr,Fe,Mg,Mn,Ni)_3O_4$ | 1000 °C | [38] | 2020 |
| Perovskite | $Ba(Zr_{0.2}Sn_{0.2}Ti_{0.2}Hf_{0.2}Me_{0.2})O_3$, Me = $Y^{3+}$, $Nb^{5+}$, $Ta^{5+}$, $V^{5+}$, $Mo^{6+}$, $W^{6+}$ | 750 °C | [47] | 2018 |
| | $(Na_{0.2}Bi_{0.2}Ba_{0.2}Sr_{0.2}Ca_{0.2})TiO_3$ | 1220 °C | [29] | 2019 |
| | $La(CrMnFeCoNi)O_3$ | 735 °C | [15] | 2020 |
| | $(La_{0.2}Pr_{0.2}Nd_{0.2}Sm_{0.2}Eu_{0.2})NiO_3$ | 635 °C | [48] | 2020 |
| | $BaSr(ZrHfTi)O_3$ | - | [49] | 2020 |
| | $BaSrBi(ZrHfTiFe)O_3$ | - | [49] | 2020 |
| | $Ru/BaSrBi(ZrHfTiFe)O_3$ | - | [49] | 2020 |
| Fluorite | $(Ce,La,Pr,Sm,Y)O$ | 1150 °C | [50] | 2017 |
| | $(Ce,Gd,La,Pr,Sm,Y)O$ | 1150 °C | [50] | 2017 |
| | $(Ce,La,Nd,Pr,Sm,Y)O$ | 1150 °C | [50] | 2017 |
| | $(Hf_{0.2}Zr_{0.2}Ce_{0.2})(Y_{0.2}Yb_{0.2})O_{2-\delta}$ | 1800 °C | [51] | 2018 |
| | $(Hf_{0.2}Zr_{0.2}Ce_{0.2})(Yb_{0.2}Gd_{0.2})O_{2-\delta}$ | 1800 °C | [51] | 2018 |
| | $(Ce_{0.2}Zr_{0.2}Hf_{0.2}Sn_{0.2}Ti_{0.2})O_2$ | 1500 °C | [40] | 2018 |
| | $(Gd_{0.2}La_{0.2}Y_{0.2}Hf_{0.2}Zr_{0.2})O_2$ | 80 °C | [41] | 2019 |
| | $(Gd_{0.2}La_{0.2}Ce_{0.2}Hf_{0.2}Zr_{0.2})O_2$ | 80 °C | [41] | 2019 |
| | $(Sc_{0.2}Ce_{0.2}Pr_{0.2}Gd_{0.2}Ho_{0.2})_2O_{3\pm\delta}$ | 800 °C | [52] | 2019 |

### 3.2. Electrical Properties

The different magnitude of lattice stress on an atom causes the lattice distortion of HEOs. In the solid solution phase, each atom is surrounded by different neighboring atoms with diverse sizes, and thus, they exert different lattice stresses and result in different strains. Compared with conventional oxides, where most adjacent atoms are of the same or similar species, the asymmetry and high structural anomaly of HEOs are obvious. We know that electron waves formed by the carrier moving in the semiconductor will be scattered, and the lattice distortion increases the probability of collision between carriers and lattice atoms, leading to greater scattering and reduction of mobility. The lower the mobility, the lower the corresponding conductivity, which means that the carrier flow is hindered and accumulates, so better dielectric performance is possible. While Berardan et al. [32] explored the special influence of copper (Cu) on distortion and disorder of the rock salt structure, they found that distortion and disorder of the crystal structure increased with the increase in measurement time, along with the monotonically raising resistivity. In addition, we know that other structural changes, such as oxygen vacancies could also affect electrical properties by alternating the ability to confine electrons outside the nucleus, electron polarizability and the probability of carrier scattering [53]. The formation of oxygen defects is due to the maintenance of charge balance. When an element with lower valence dope into the non-transition metal system, it has to form oxygen vacancies to compensate the lowering charge. However, since the transition metals have multiple oxidation states, the compensation can be completed by the valence change of transition metals [54]. Therefore, it might

lead to the inhibition of oxygen vacancies and has great potential in device application. Besides, oxygen defects could be regulated through control of oxygen partial pressure. By investigating the relationship between diffusion coefficient and oxygen partial pressure during the preparation of $(Co,Cr,Fe,Mn,Ni)_3O_{4-y}$, Grzesik et al. [55] concluded that oxygen content of the final product would be reduced at low oxygen partial pressure levels, implying that corresponding oxygen vacancy concentration would be higher than at high oxygen partial pressure. So, by means of controlling oxygen vacancies, we could regulate the devices' electrical performance [56].

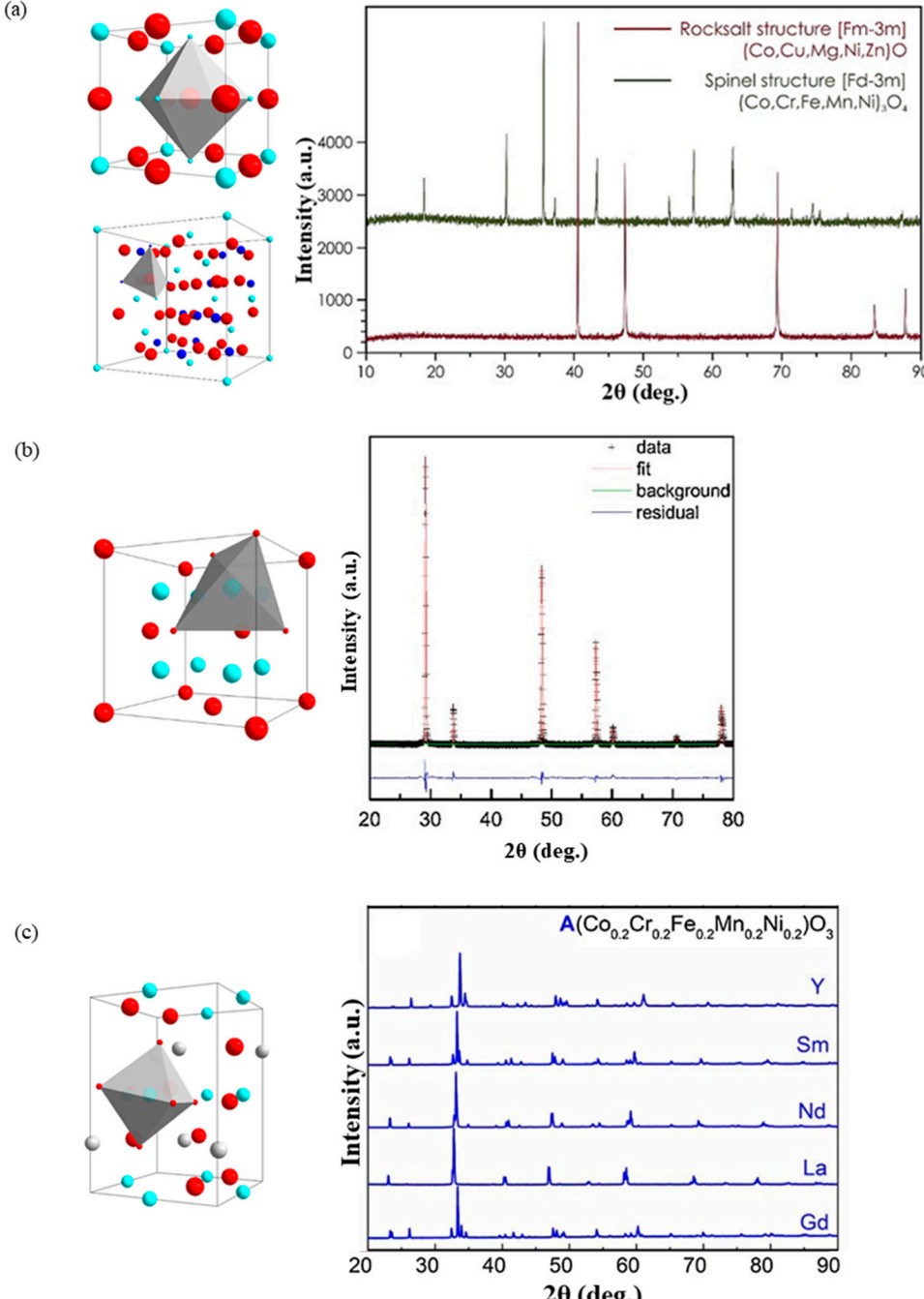

**Figure 4.** Four main HEOs' crystal structures along with X-ray diffraction patterns of the representative materials: (**a**) rock salt and spinel [44]; (**b**) fluorite [40]; (**c**) perovskite [39].

The best-known research concerning the dielectric property of high-entropy oxides is that of David Bérardan and his fellows [13], who researched the doping effects on HEOs' dielectric constant. After trying to dope (Mg, Ni, Co, Cu, Zn)O with indium(In), gallium(Ga), titanium(Ti), and lithium(Li), respectively, they found that the doping of other elements except Li would lead to the formation of non-single-phase crystals. Since the radius of $In^{3+}$ is larger than that of the original five-element metal cation, and the 2+ metal cation's radius is larger than $Ga^{3+}$, it can be concluded that the second phase is independent of the ion radius. They speculated that this was because most of the 3+ or 4+ cations in binary oxides existed stably in octahedral configuration and were not easily converted to the cubic close packing like cations in (Mg, Ni, Co, Cu, Zn)O. On the contrary, the crystal structure of lithium dioxide is anti-fluorite, which means it is easy to replace the cationic sites in (Mg, Ni, Co, Cu, Zn)O. This is also supported by the lattice coefficient. Because $Li^+$ is monovalent and its valence state is lower than other cations, in order to maintain charge balance, it is necessary to compensate the charge difference through oxygen defects or oxidation of $Co^{2+}$ into $Co^{3+}$. Then, p-type doping will lead to a large number of carrier concentrations or impurity bands close to the maximum value of the valence band. For all these $HEO_X$ samples, the resistance changes exponentially with temperature, just as the gap decreases with the substitution of Li in a semiconductor with an about 1 eV bandgap. This newly discovered material has a large dielectric constant and two kinds of dielectric relaxation, and the lower the lithium content, the higher the dielectric constant (above 4800 in the measured temperature range [13] (Figure 5)).

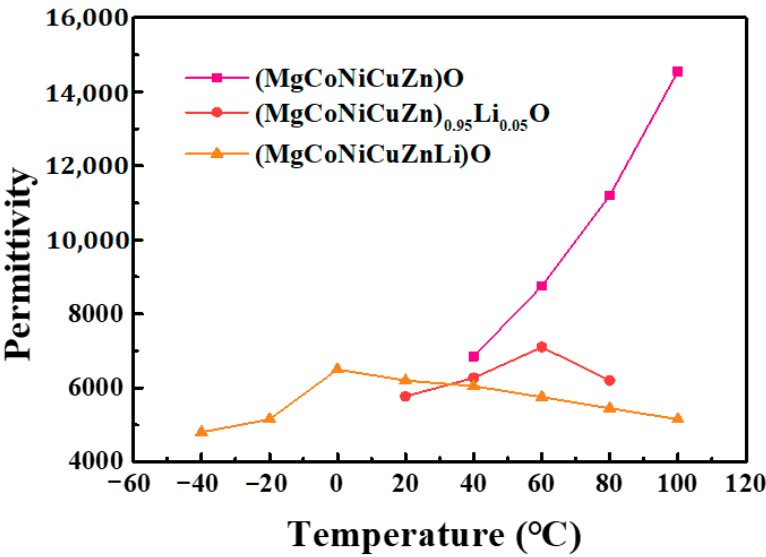

**Figure 5.** Permittivity of samples under different temperatures [13].

Pu et al. [29] also reported the dielectric relaxation of HEOs. Through Curie–Weiss Law (Formula (3)):

$$\frac{1}{\varepsilon_r} - \frac{1}{\varepsilon_m} = \frac{(T - T_m)^\gamma}{C} \tag{3}$$

where $\varepsilon_m$ refers to the maximum dielectric constant, $T_m$ is the corresponding temperature value under the maximum dielectric constant, $C$ is the Curie–Weiss constant, $\gamma$ indicates the degree of dielectric relaxation. They determined that the $\gamma$ of perovskite $(Na_{0.2}Bi_{0.2}Ba_{0.2}Sr_{0.2}Ca_{0.2})TiO_3$ equals 1.72, very close to the ideal relaxation ferroelectrics. In typical relaxation ferroelectrics, a structural disorder caused by ion doping results in the relaxation, so it is reasonable to assume that the more diverse elements fill in, the more distorted structure is, which is the reason for relaxation in HEOs.

In addition, an experiment by Shiyu Zhou et al. [57] provides a reasonable explanation for why high-entropy oxides have huge dielectric constants. Figure 6 shows

a low dielectric loss and excellent stability within a certain temperature range while testing dielectric properties of perovskite structure Ba $(Zr_{0.2}Ti_{0.2}Sn_{0.2}Hf_{0.2}Nb_{0.2})O_3$ and Ba$(Zr_{0.2}Ti_{0.2}Sn_{0.2}Hf_{0.2}Ta_{0.2})O_3$. By testing the complex impedance spectra and the simulation of the parallel R–C equivalent circuit of the two groups of materials, they found that the grain boundary resistance ($R_{gb}$) was lower than the grain resistance ($R_g$). Meanwhile, according to the Arrhenius Formula (4), where k is the Boltzmann constant estimation:

$$\sigma = \sigma_0 e^{-\frac{E_a}{kT}} \tag{4}$$

the activation energy of grain boundary ($E_{gb}$) and activation energy of grain ($E_g$) have an obvious difference (for Ba$(Zr_{0.2}Ti_{0.2}Sn_{0.2}Hf_{0.2}Nb_{0.2})O_3$, $E_g$ = 0.68 eV, $E_{gb}$ = 0.70 eV; for Ba$(Zr_{0.2}Ti_{0.2}Sn_{0.2}Hf_{0.2}Ta_{0.2})O_3$, $E_g$ = 0.59 eV, $E_{gb}$ = 0.68 eV), meaning that severe lattice distortion caused by various cationic complex arrangements and a highly disordered grain internal lead to the failure of electronic gather in the grain boundary and movement transmission over long distances. This means that it has a greater possibility of storing charges, which is the performance of the dielectric property. Therefore, it can be seen that the high-entropy structure plays an important role in the dielectric constant stability and the low dielectric loss (tan δ < 0.002).

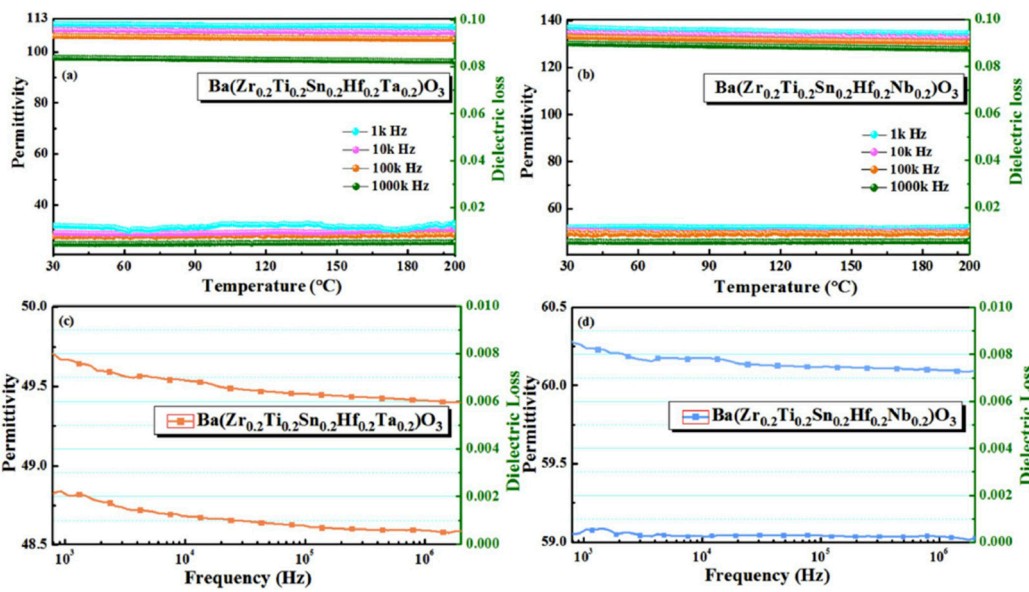

**Figure 6.** Performance of permittivity and dielectric loss versus (**a**,**b**) temperature and (**c**,**d**) frequency [57].

## 4. High-Entropy Oxide Thin Films

### 4.1. Preparation Methods

Most present studies are about high-entropy ceramic materials or high-entropy powder materials, but few studies are about thin films. Existing preparation methods HEO films are all physical methods. The most common one is the sputtering deposit because it is suitable for any material, including high-melting-point metals, where HEOs usually require a high temperature to form. Some scientists deposit metals by magnetron sputtering [5], radio frequency (RF) sputtering [58] or reactive sputtering [28,59] under vacuum first, and then convert alloy films into oxide films via high-temperature annealing with the presence of oxygen. In addition, vacuum evaporation is also a good choice. Jacobson et al. [60] prepared polycrystalline HEO thin films by pulsed laser deposition (PLD) using MgO, CoO, NiO, CuO, ZnO, and this method guarantee components stoichiometric ratio.

Except methods mentioned above, some scientists have also attempted to use solution-based methods to derive HEO films. Chen et al. [61] successfully obtained the homogeneous Ba(Ti,Hf,Zr,Fe,Sn)O$_3$ thin film by spin coating and hydrothermal synthesis. According to

the report by Liang et al. [62], they got the $Ba(Zr_{0.2}Sn_{0.2}Ti_{0.2}Hf_{0.2}Nb_{0.2})O_3$ film even just by the sol-gel method. So, perhaps solution-based methods could become the potential one for preparing HEO thin films, because of their low-temperature requirements, simple operation, and great industrial production prospects. Still, scientists could try out some new preparation methods, such as ink-jet printing or screen-printing, in order to improve thin films' properties and morphology. Screen-printing and inkjet printing technology can effectively do subtraction to the fabrication process of electronic products. They eliminate the evaporation with the use of a mask and subsequent photoetching, greatly reducing production costs. This lay the foundation for HEO thin films to be really put into large-scale industrial production finally.

### 4.2. Optoelectronic Properties

Jacobson et.al. [60] found that deposition temperature and oxygen pressure have a great influence on $(Mg_{0.2}Co_{0.2}Ni_{0.2}Cu_{0.2}Zn_{0.2})O$ film resistivity. The measurement result of the high temperature low pressure (HTLP) sample was $3.42 \pm 0.175$ MΩ cm nonlinear resistivity, while that of the high temperature high pressure (HTHP) sample shows that nonlinear resistivity was $1.2 \pm 0.05$ MΩ cm, and the results for the low temperature low pressure (LTLP) sample resistivity are close to linear, about $51 \pm 0.65$ kΩ cm. In contrast, the resistivity of LTLP increases 20 times by sufficient low temperature annealing under an oxygen-rich environment, approaching that of HTHP and HTLP samples, while there is also an increase in the lattice constant. Therefore, they speculate that the conductive mechanism is polaron jumping since previous studies found that low-temperature sediments tend to transform from $Co^{2+}$ to $Co^{3+}$ and cause lattice shrinkage. Thus, polarons in the compressed lattice have a shorter jumping distance, which makes jumping more likely to occur and reduces resistivity [58,63].

The oxygen vacancies have similar effects on HEO thin films as on HEOs. Miao-I Lin [5] found that the oxygen content astonishingly affected resistance when testing $(AlCrTaTiZr)O_x$ thin film resistivity—the resistivity of the metal film is about $10^2$ μΩ cm, but when oxygen concentration increases, the resistivity rises sharply to $5 \times 10^{11}$ μΩ cm. Furthermore, Huang et al. [58] measured the resistivity of the high-entropy oxide thin film constituted with aluminum (Al), chromium (Cr), iron (Fe), cobalt (Co), nickel (Ni), and Cu six elements. The research indicated that 20% oxygen content is an inflection point (0–20%: resistivity increases with oxygen content; >20%: the trend is opposite, but the lowest resistivity is still greater than the pure high-entropy alloy $(AlCrFeCoNiCu_{0.5})$ film resistivity). This is because metal vacancies increase with excess oxygen in the semiconductor, which indicates that the conductivity of the oxide film depends on the metal vacancies. In addition, annealing temperature has fewer effects on the grain size but will enlarge the micropore size between grains, paving a way to larger resistivity.

Tsau et al. [63] tested the lowest resistivity under different annealing conditions with different components and found that the resistivity increased with the total number of element types, as shown in Table 2. Specifically, for $Ti_xFeCoNiO_y$, the resistivity rises with the increase in Ti content. Although $Ti_xFeCoNiO_y$ and $FeCoNiO_y$ possess lower resistivity (~30 Ω cm), technically, they are not real high-entropy oxide. Therefore, we can see that the influence of high entropy on film resistivity is remarkable and significant.

**Table 2.** Lowest resistivity for samples with different compositions under their own best annealing conditions.

| Sample | Lowest Resistivity ($\mu\Omega$ cm) | Sputtering Power | Annealing Temperature |
|---|---|---|---|
| $FeCoNiO_y$ | 19 | | 1000 °C |
| $Ti_{0.25}FeCoNiO_y$ | 28 | | 1000 °C |
| $Ti_{0.5}FeCoNiO_y$ | 26 | | 1000 °C |
| $Ti_{0.75}FeCoNiO_y$ | 31 | | 900 °C |
| $TiFeCoNiO_y$ | 35 | 100 W | 1000 °C |
| $TiFeCoNiCuO_y$ | 130 | | 450 °C |
| $TiFeCoNiCu_2O_y$ | 250 | | 350 °C |
| $TiFeCoNiCu_3O_y$ | 101 | | 400 °C |
| $Al_{0.5}CrFeCoNiCuO_y$ | 550 | | 600 °C |
| $AlCrFeCoNiCuO_y$ | 1500 | | 500 °C |

In addition to resistance, HEO thin films might also have applications in semiconductors. Chen et al. [59] tested those same six-elemental oxides. Hall measurements show $Al_{0.5}CoCrCuFeNi$ oxide film with the P-type conducting behavior, the conductivity of which is 40.1 $(\Omega$ cm$)^{-1}$ (i.e., 0.0249 $\Omega$ cm resistivity), carrier density is $5.81 \times 10^{18}$ cm$^{-3}$, and mobility reaches up to 43.2 cm$^2$V$^{-1}$s$^{-1}$. As the Al content increases, the number of conductive cations at the octahedral sites drops down; this also applies to available carriers and mobility, which explains the decrease in the film conductivity. When it comes to the N-type semiconductor of HEOs, the representative is $(ZnSnCuTiNb)_{1-x}O_x$ prepared by Yu's team [28] through sputtering. They originally want to discover the relationship between oxygen levels (50.3%, 51.6%, 56%, 59.2%) and their respective performances but found that only two types of oxide film (51.6% and 56%) have characteristics of optoelectronic semiconductors. Additionally, their carrier concentrations are $2.62 \times 10^{20}$ cm$^{-3}$ and $1.37 \times 10^{17}$ cm$^{-3}$, and their electrical conductivities are 57.2 $(\Omega$ cm$)^{-1}$ and $9.45 \times 10^{-3}$ $(\Omega$ cm$)^{-1}$, with indirect band gaps of 1.69 eV and 2.26 eV, respectively. Compared with common n-type doped $SnO_2$ thin film (carrier concentrations ~$10^{20}$ cm$^{-3}$) [64] and p-type doped ZnO thin film (carrier concentrations ~$10^{17}$ cm$^{-3}$) [65], it seems that HEO thin films have a great potential in semiconductor applications with up to $10^{20}$ cm$^{-3}$ carrier concentrations.

Besides electrical properties, some scientists reported HEO thin films' optical properties as well. Lin et al. [5] found that oxygen content also influences films' optical transparency apart from resistivity. As the percentage of oxygen element grows from 0% to 50%, the film becomes transparent from being originally opaque. They contributed it to the type of bonding, that is, oxygen atoms occupy interstitial sites of solid solution at 2.5%, while at 15% ionic bonds and bandgap form. This explanation has been confirmed by Yu et al. [28], who discovered that the transparency went from 10% to 50% with only 8 atom.% difference in oxygen (Figure 7). Furthermore, Chen et al. [59] reported that aluminum has a similar behavior as oxygen (Figure 7), which enlarges indirect bandgap from 1.45 to 1.56 eV and direct bandgap from 2.15 to 2.49 eV as aluminum molar ratio increases from 0.5 to 2.0. A higher bandgap means a higher energy is needed to excite electrons; if the incident light energy is not satisfied, it will not be absorbed. Thus, more visible light passes through, which appears transparent.

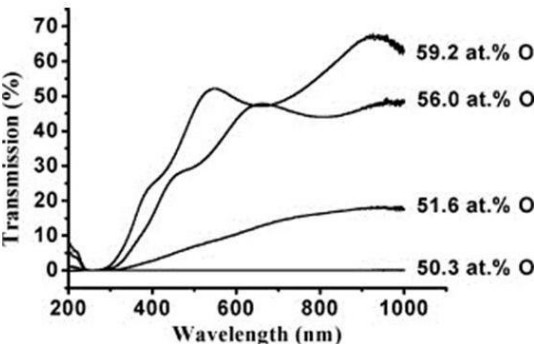

**Figure 7.** Optical transmission with different parameters: oxygen content [28].

Interestingly, so far, no research has reported the excellent dielectric properties of HEO thin films compared with the colossal dielectric constant discovered in the HEO. We speculated that this has a relationship with the structure of HEO thin films. HEOs with a large dielectric constant are either the perovskite structure or the rock salt one, that is, in a word they have a crystal structure. Additionally, lattice distortion brought by high-entropy effects plays a significant role in electronic gather in the grain boundary and movement transmission over long distances. Additionally, thus, it results in good dielectric properties in HEOs. However, many HEO thin films in existing reports are amorphous, for instance, Lin et al. [5] and Yu et al. [28], which have low resistivity, compared with Jacobson group's [60] polycrystalline with large resistivity. Second, elements also potentially have some influence on it. For example, Cu, Al, Fe, Zn, etc., are famous conductive elements. Therefore, it is no wonder that $(ZnSnCuTiNb)_{1-x}O_x$ and $Al_{0.5}CoCrCuFeNi$ even possess semiconductor properties. If we want to get good dielectric properties, which is on the basis of HEO thin films' high resistivity, then properly increasing oxygen content, temperature and pressure may be possible. Furthermore, scientists can find some new preparation methods in order to change the structure and properties of HEO thin films fundamentally. Like we mentioned above, Chen et al. [62] prepared HEO films at 200 °C using the hydrothermal synthesis method, which is a good example. Water, as a pressure transfer medium, realizes the formation and improvement of inorganic compounds by accelerating the osmotic reaction and controlling the physical and chemical factors of its process. Hydrothermal synthesis uses high-pressure aqueous solutions to dissolve and recrystallize substances that are insoluble under atmospheric conditions. Therefore, it can replace some high-temperature solid-state reactions and significantly reduce the reaction temperature by 100–200 °C. We concluded that this fits in with the idea of increasing the resistivity by increasing the pressure. At the same time, wearable devices have become the inevitable trend for electronic products, and the substrate will mostly be the organic one instead of the traditional silicon, so the processing temperature of the devices should not be too high.

## 5. Prospects and Applications

### 5.1. Existing Mature Applications

Due to solid-solution strengthening, high-entropy oxides display high hardness. Additionally, thanks to slow diffusion limiting the penetration of oxygen or other substances, they also possess good oxidation resistance and corrosion resistance. Nowadays, a large number of studies have already measured the hardness, elastic modulus, and other mechanical properties of different kinds of HEOs. Among them, the hardness of $Al_2(CoCrCuFeNi)O$ [66] and $(AlCrTaTiZr)O$ [5,66] films are the highest reported in the existing oxide coatings, with $22.6 \pm 1.5$ GPa and 20 GPa, respectively. They have high application value by being used as protective coatings for mechanical parts, such as cutting tools and drills, which usually work under high-temperature corrosive environments (Figure 8a) [67].

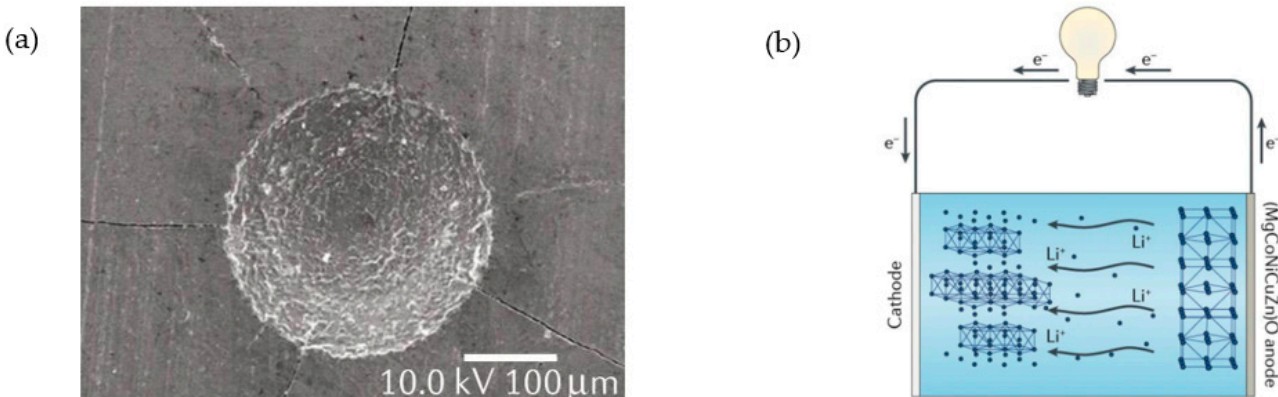

**Figure 8.** Applications of HEOs such as (**a**) wear-resistant coatings and (**b**) rechargeable batteries [68].

Apart from that, there are many pieces of research concerning HEOs' electrical properties as well. Li-doped HEO exhibits extremely high lithium-ion conductivity at room temperature, covering a few orders of magnitude compared with conventional solid electrolytes such as lithium phosphorousoxynitride (LiPON), and is a good alternative to anode materials for future lithium-ion batteries [14,67–69] (Figure 8b).

### 5.2. Future Candidate for Dielectric Materials in TFT

Apart from existing applications, combined with the huge dielectric constant discovered in HEOs and high-entropy theory, it is reasonable to look forward to its future application in electronic devices. This is because, currently, a high dielectric constant may effectively solve the problem of tunneling current that microelectronic devices bring along with them. For example, besides supercapacitors, replacing the dielectric layer in thin-film transistors (TFT) is certainly one of its possible applications. TFT plays a vital role in the display; it drives each pixel on the display. Thin-film transistor liquid crystal display (TFT-LCD) is one of the best color displays because of high responsivity, high luminance, high contrast ratio, and other advantages.

Dielectric materials of the TFT dielectric layer are electrical insulators, which become polarized when an electric field is applied. The positive gate voltage ($V_G$) electrostatically induces gate dielectric to polarize, making electrons accumulated on the surface between the semiconductor layer and the dielectric layer. The drain voltage ($V_d$) is applied and the drain current ($I_d$) is maintained in the channel layer between the metal source and the drain [70] (Formulas (5) and (6)):

$$V_D \leq V_G - V_T : \ I_D = \frac{W}{L}\mu C \left\{ (V_G - V_T) - \frac{V_D}{2} \right\} V_D \tag{5}$$

$$V_D \geq V_G - V_T : \ I_{D,sat} = \frac{W}{L}\mu C (V_G - V_T)^2 \tag{6}$$

When $V_G$ is less than $V_T$, there is a leakage current between the source and drain, described by the subthreshold swing ($SS$) [71] (Formula (7)):

$$SS = \frac{kTln10}{q}\left(\frac{q^2 N_b}{C} + 1\right) \tag{7}$$

where $k$ is the Boltzmann constant, $q$ is the elementary charge, $T$ and $C$ are the abbreviations of temperature and capacitance, respectively, and $N_b$ represents the total well density.

In order to improve mobility of the carrier, we should reduce defect positions that can prevent effective polarization as much as possible. Low surface roughness has a positive effect on these defects and further enhances the accumulation of charge. Other

requirements, such as high film density to reduce leakage current and lower film thickness to increase dielectric constant, also benefit improving carrier mobility [72].

Silicon dioxide ($SiO_2$) is the best option for gate dielectrics. However, the leakage current of it will increase sharply along with the shrinking size of traditional $SiO_2$ dielectric films, which contradicts with the trend of electronic components miniaturization and integration. Studies [73] have shown that the $SiO_2$ channel with a physical thickness less than 7Å overlaps with the silicon-rich interface of the polycrystalline silicon gate interface in MOSFET, resulting in an effective "short circuit" of the dielectric, making it unable to function as an insulator. However, the high dielectric constant(high-k) material can not only guarantee enough driving current but also increase the effective thickness of the oxide layer while keeping the equivalent oxide layer physical thickness, which inhibits the quantum tunneling effect. In fact, apart from the traditional group III (such as $Al_2O_3$) and group IV ($ZrO_2$) etc., more and more scientists [74,75] conduct surveys by starting trying binary or ternary oxides, and essentially this is gradually heading towards the "high entropy" direction. Therefore, it is possible for great dielectric constant HEOs to be one of the solutions to the problem.

Another reason supporting this future application is the slow diffusion rate. Nowadays, the developmental trend of electronic equipment is becoming smaller and thinner, followed by the increase of tunneling current. Therefore, it is necessary to find new materials to act as a diffusion barrier between the substrate and the active layer. The crystal structure can greatly affect the speed of diffusion. Generally speaking, the diffusion rate in a close-packed structure is slower than that in a non-close-packed structure. And HEOs' cations or anions always exist in close-packing arrangements such as spinel and fluorite. Lattice distortion caused by different element radii, together with increased bulk density, lead to slow diffusion even in only a few- nanometers-thick layers68. Experiment by Grzesik et al. [55] confirmed that the chemical diffusion coefficient (D) of oxygen in (Co, Cr, Fe, Mn, Ni)$_3O_4$ was within the typical range of transition metal oxides at high $O_2$ pressure, while at low oxygen partial pressure, significantly lower D may indicate a strong interaction between defects, that is, this is due to the complexity of the defect structure in the high entropy spinel. And this further enhances the feasibility and great potential of high entropy oxide films acting as the dielectric layer in TFT.

In the solid solution, the diffusional activation energy of substitutive atoms is larger than that of interstitial atoms, because vacancies have to be formed first and then carry out diffusion. Many experiments have already proved that most of HEOs' atoms enter the crystal structure by displacement rather than interstitial filling, so it harms the diffusion rate. At the same time, if the crystal contains allotropic metals, such as Fe/Co/Ti/Sn/Mn, the diffusion rate would also be affected.

At present, relatively mature studies still concentrate on high-entropy nitrides. For instance, a 4 nm thick alternating structure with the hex-elemental nitride $(AlCrRuTaTiZr)N_{0.5}$ and the hexatomic alloy AlCrRuTaTiZr, is an advanced solid diffusion barrier layer. In the process of high-temperature (800 °C) annealing, this barrier effectively retards the mutual diffusion of Cu and Si so that no Cu silicide forms.

## 6. Conclusions

The oxides' properties and performances are closely related to their structures. High entropy causes lattice distortion and the elemental synergistic effect, which makes HEOs have remarkable characteristics, such as high hardness, wear resistance, outstanding stability of entropy, large dielectric constant, and good electrical conductivity, etc. So, it is necessary to study this further so that we can determine the full potential of these materials. While exploring the diversity of properties, there is a need to further clarify the influencing factors and principles behind them, because only in this way can we achieve controllability in the design of materials.

There is a wide range of potentials in terms of distinctive properties, including energy storage, catalysis, materials for electronic devices, and wear-resistant and biocompatible

coatings. At present, the research of HEOs as protective coatings is extensive, but the application and development of their electrical properties are relatively backward. Interestingly, the large permittivity of the (MgCoNiCuZn)O rock salt crystal family that has been discovered so far may be a supporting material for HEOs to be a strong alternative to the dielectric layer materials of TFTs in the future. Although until now the dielectric property has not been reported in HEO thin films, we can reasonably deduce that it is a corollary that HEO thin films are able to possess the dielectric property since the structural lattice distortion effect results in greater scattering and reduction of carrier mobility. In agreement with present studies, increasing oxygen content, temperature and pressure may be effective to enhance its dielectric property because it influences resistivity. Alternatively, scientists could try to solve this problem fundamentally by, e.g., changing a new thin-film preparation method so that it can guarantee the formation of a crystal structure to magnify the high-entropy effect or changing element composition to change the cocktail effect. Thus, we anticipate finding its application in TFT dielectric films.

**Author Contributions:** Conceptualization, H.L. and Y.Z.; methodology, Y.Z. and Z.L.; software, H.N., R.Y. and J.P.; validation, T.Q. and W.X.; formal analysis, H.L., X.F. and Z.X.; investigation, Y.Z. and Z.L.; resources, H.N., R.Y. and J.P.; data curation, H.L., X.F. and Z.X.; writing—original draft preparation, H.L.; writing—review and editing, H.L., Z.L. and H.N.; visualization, T.Q. and W.X.; supervision, H.N., R.Y. and J.P.; project administration, H.L.; funding acquisition, X.F. All authors have read and agreed to the published version of the manuscript.

**Funding:** This work was supported by Key-Area Research and Development Program of Guangdong Province (No.2019B010924001), National Natural Science Foundation of China (Grant No. 62074059 and 22090024), Guangdong Major Project of Basic and Applied Basic Re-search (No.2019B030302007), Guangdong Basic and Applied Basic Research Foundation (No.2020B1515120020), Fundamental Research Funds for the Central Universities (No.2020ZYGXZR060 and 2019MS012), Ji Hua Laboratory scientific research project (X190221TF191), National College Students' Innovation and Entrepreneurship Training Program (No.202010561001, 202010561004 and 202010561009), South China University of Technology 100 Step Ladder Climbing Plan Research Project (No.j2tw202102000) and 2021 Guangdong University Student Science and Technology Innovation Special Fund ("Climbing Plan" Special Fund) (No.pdjh2021b0036).

**Institutional Review Board Statement:** Not Applicable.

**Informed Consent Statement:** Not Applicable.

**Data Availability Statement:** Data is available on request from the corresponding author.

**Conflicts of Interest:** The authors declare no conflict of interest.

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
