# Peer review of "High-Entropy Oxides: Advanced Research on Electrical Properties"

_coatings, doi:10.3390/coatings11060628_

Round 1

Reviewer 1 Report

Manuscript  ID: coatings-1215137

Manuscript Title:” High entropy oxides (Bulk or Film): Advanced research on electronic properties”

Authors: Haoyang Li, Yue Zhou, Zhihao Liang, Honglong Ning *, Xiao Fu,
Zhuohui Xu, Wei Xu, Tian Qiu, Rihui Yao *, Peng Junbiao”

Recommendation: Accept after minor revision

Additional comments: The target of the manuscript is to represent the review about electronic properties multi-component metal oxides (so called High Entropy Oxides (HEOs)). This target is important and actual. So, the manuscript is dedicated to an actual problem of physical chemistry, metallurgy and electronics.

However, there are some small problems in the manuscript that should be corrected.

  1. Figure numbers should correspond to the respective numbers in the text of the paper.

  1. As it was mentioned in the paper “However, the fabrication temperature of HEOs is usually high, and the mechanism behind it is not clear yet, so it is necessary to do more researches in order to effectively guide the design and preparation of materials.” So in the Table 1 it will be very useful to show the column with the preparation temperature for the respective compounds.

  1. Instead the Figure 4, that is too schematic it can be interesting to see some crystal structures that correspond to the compounds mentioned in the paper along with their XRD patterns.

  1. It should be also mentioned that during thin film preparation the temperature of the substrate is normally much less than it is necessary for the kinetic mobility of the components can provide equilibrium. So the post preparation annealing is necessary for oxidation and stabilization of the crystal structure of the compounds.

          In the Table 2 preparation temperatures and annealing temperatures              can be represented in the separate columns.

Author Response

Dear Reviewer: 

Thank you for your letter and the comments concerning our manuscript.  Those comments are meaningful and  helpful for revising and improving our paper. We have studied comments carefully and have made corrections, which we hope meet with approval.  For detailed information, please see the attachment.

Please infrom us if you have any questions.

Reviewer 2 Report

The manuscript "High Entropy Oxides (Bulk or Film): Advanced Research on Electronic Properties" by Haoyang Li et al. deals with the basic explanation of high-Entropy, journal classification of HEOs based on their structures, electrical properties of HEO with film, and study of their potential applications. The topic seems interesting and actual, but I do have some suggestions as to the content of the manuscript before finalizing the publication of this manuscript in Coatings.

To begin with, (a) the considerable mistakes of typos and missing texts are observed in their write-up, for example, In figure caption (1) they used different numbering in the main text and caption, Same mistakes repeated in the remaining portion of the manuscript, Please use uniform format throughout the manuscript.

(b) Is it a great idea to use a short form of ‘have not’ in scientific report writings! (see your abstract)

(c) While writing the author’s name, they mentioned sometimes the first author’s name or the second author’s name, etc., and skipped ‘et. al’. Please use a uniform sequence/pattern in the writing.

(d) There are mistakes in the correct use of English grammar, therefore, it is needed to address the linguistic skills at least for this write-up. (for example, check the introduction part)

(e) Please correct the equation numbering!

(f) Please specify the atoms/molecules type in figure 330 of lattice crystal structure or reorganize the figure caption to make the explanation clearer.

(g) Add the most recent article for reference at the places of 29th and 30th articles.

(h) Please refer to some recent articles in the part of 3.2 ‘electrical properties’ to make the explanation clearer about the oxygen defects in transition metals, such as Co. The following article has a reasonable explanation for device application. It can be helpful for the readers if you include it in the reference list.

"Ajmal, H.M.S.; Khan, F.; Nam, K.; Kim, H.Y.; Kim, S.D. Ultraviolet photodetection based on high-performance Co-plus-Ni doped ZnO nanorods grown by hydrothermal method on a transparent plastic substrate. Nanomaterials 2020, 10, 1–20"

(i) In figure 7, there are a lot of corrections required as in other figures are demanded, I mentioned this for the sake of example. Why the Y-axis is different in each? one is transmission while the other is transmittance, why the wavelength ranges in both figures are not the same? Do both figures have different formats?

Please revise your manuscript carefully and read it many times before the next submission.

Check your reference style according to the guidelines of Journal ‘Coatings’

Author Response

Dear Reviewer: 

Thank you for your letter and for the reviewers’ comments concerning our manuscript. Those comments are valuable and very helpful for revising and improving our paper. We have studied comments carefully and have made correction which we hope meet with approval. For detailed information, please see the attachment. And about the reference format, we have adjusted in the manuscript.

Please kindly let us know if you have any questions.

Reviewer 3 Report

Referee Report

on paper “ High Entropy Oxides (Bulk or Film): Advanced Research on Electronic Properties“ (coatings-1215137) by authors  Haoyang Li1, Yue Zhou, Zhihao Liang, Honglong Ning, Xiao Fu, Zhuohui Xu, Tian Qiu, Wei Xu, Rihui Yao and Junbiao Peng submitted to Coatings

This is interesting paper. It reports about the correlation between the chemical composition, entropy state and functional properties of the high-entropy oxides (HEOs). HEOs nowadays are very importand and attractive class of materials. The presented results are interesting and reliable. However, paper needs some improvement only after which it can be accepted. At this stage, my decision is major revision. But I hope that after revision this paper can be accepted. I impressed by the paper.

  1. Title: I feel that text in brackets (Bulk or Film) can be deleted. Also after careful evaluation of the text I feel that there is no any ELECTRONIC properties. I observed only ELECTRICAL properties. Electronic and electrical are different properties. Thus I thing that title “High Entropy Oxides: Advanced Research on Electrical Properties” seems more general and logical.
  2. Abstract written well. But I recommend highlight most important points. Please describe practical importance.
  3. I fully agree with the author that HEOs attract great attention due to strong correlation between chemical composition, configuration entropy and many physical properties. They are attractive from fundamental point of view and perspective for practical applications. But I feel that in introduction it can be discussed scientific importance of the HEOs based on magnetic transition metal ions (DOI: 10.3390/nano11041014, DOI: 10.1016/j.ceramint.2021.03.088, DOI: 10.1016/j.jeurceramsoc.2020.04.036).
  4. Something strange in figure numbering? Why Fig. 1 has No. 11, and Fig. 2 has number 224, Fig. 3 – 330 etc.? Please check everything carefully and revise this.
  5. There are a lot of typos and mistakes in the text. For example – page 7 line 3 after equation (4) – Eg must be in eV (in text in V)

The paper should be sent to me for the second analysis after the major revisions.

Author Response

Dear Reviewer: 

Thank you for your letter and for the reviewers’ comments concerning our manuscript. Those comments are meaningful and helpful for revising and improving our paper. We have studied comments carefully and have made correction which we hope meet with approval. For detailed information, please see the attachment.

Please kindly let us know if you have any questions.

Round 2

Reviewer 2 Report

The authors have improved the manuscript accordingly. However, a minor spell check and editing are required before publication. 

Reviewer 3 Report

Authors made brutal revision in accordance with my comments. I feel that revised version can be accepted in present form.